# Perceptions of the Challenges and Opportunities of Utilising Organic Waste through Urban Agriculture in the Durban South Basin

**DOI:** 10.3390/ijerph17041158

**Published:** 2020-02-12

**Authors:** Nqubeko Neville Menyuka, Melusi Sibanda, Urmilla Bob

**Affiliations:** 1Department of Agriculture, Faculty of Science & Agriculture, University of Zululand, KwaDlangezwa 3886, South Africa; nqubekomenyuka501@gmail.com; 2Department of Geography, School of Agriculture, Earth & Environmental Science, University of KwaZulu-Natal, Westville 54001, South Africa; Bobu@ukzn.ac.za

**Keywords:** eThekwini Municipality, composting, recycling of solid waste, sustainable agriculture

## Abstract

Waste management has become pertinent in urban regions, along with rapid population growth. The current ways of managing waste, such as refuse collection and recycling, are failing to minimise waste in cities. With urban populations growing worldwide, there is the challenge of increased pressure to import food from rural areas. Urban agriculture not only presents an opportunity to explore other means of sustainable food production, but for managing organic waste in cities. However, this opportunity is not taken advantage of. Besides, there is a challenge of mixed reactions from urban planners and policymakers concerning the challenges and benefits presented by using organic waste in urban agriculture. The current paper explores the perceived challenges and opportunities for organic waste utilisation and management through urban agriculture in the Durban South Basin in eThekwini Municipality in KwaZulu-Natal (KZN) Province of South Africa. It is anticipated that this information will be of use to the eThekwini Municipality, policymakers, researchers, urban agriculture initiatives, households and relevant stakeholders in the study areas and similar contexts globally. Two hundred (200) households involved in any urban farming activity and ten (10) key informants (six (6) staff from the Cleaning and Solid Waste Unit of the eThekwini Municipality and four (4) from the urban agricultural initiative) were selected using convenient sampling. Descriptive statistics and inductive thematic analysis were used to analyse data. The significant perceived challenges and risks associated with the utilisation of organic waste through urban agriculture included lack of a supporting policy, climatic variation, lack of land tenure rights, soil contamination and food safety concerns. Qualitative data further showed that the difficulty in segregating waste, water scarcity, difficulty in accessing inputs, limited transportation of organic waste, inadequate handling and treatment of organic waste, and being a health hazard were some important challenges. On the other hand, the significant perceived benefits associated with the utilisation of organic waste through urban agriculture were enhanced food and nutrition security, and opportunities for business incubation. Other important benefits established through qualitative data were an improved market expansion for farmers and improved productivity. Overall, despite the perceived challenges and risks, there is an opportunity to manage organic waste through urban agriculture. It is imperative for an integrated policy encompassing the food, climate and waste management to be developed to support this strategy. All stakeholders—the government, municipal authorities and urban agricultural initiatives should also, guided by the policy, support urban farmers, for example, through pieces of training on how to properly manage and recycle organic waste, land distribution, inputs availability and water usage rights among other things.

## 1. Introduction

Waste management is now considered to be one of the critical issues commonly found in the metropolitan regions across the world. Worldwide, urban populations are increasing [1]. Rapid population growth puts pressure on available infrastructure concerning waste collection and management [2]. Waste management has become a pertinent issue globally, especially in metropolitan areas [3]. There is evidence from the literature that shows that annually, billions of tons of waste produced is from municipalities and industries [4]. Organic waste is a significant part of municipal solid waste (MSW) that is increasing due to rapid urbanisation.

Researchers have defined the concept of organic waste, as organic material such as food, garden, plant and animal-based material as well as degradable materials [3,5]. Usually, in urban areas, organic waste is disposed of by taking it to landfills [6]. This disposal method does not only present a problem of lost resources but also contributes to environmental challenges (land, air and water pollution). In the middle of the 1970s, environment specialists or environmentalists started to criticise the disposal of waste methods. The concept of the waste management hierarchy emerged as a good management strategy [3,7]. The concept of the waste management hierarchy, as shown in Figure 1, depicts all measures that take place when trying to avoid the negative impact of waste on the environment. These measures can include; reuse or recycle disposal and energy recovery. 

Generally, the key position of the waste management hierarchy is that preference should be given to those actions found at the top when compared to those at the lower level of the hierarchy [8,9]. The key motive behind this is that avoiding waste is more useful than re-use, which is more beneficial than recycling while recycling is also much better than landfilling or incineration [3,10]. Figure 1 shows the pillars of the concept of the waste management hierarchy.

The pillars of the concept of waste management incorporate the following aspects: waste prevention (reduction); re-use; recycling; energy recovery and disposal. Reduction refers to actions that happen before any material is considered waste [11]. Prevention requires a collaborative effort of manufacturers, households and other stakeholders in the economy to change behaviour towards waste prevention process. Although some legislation may exist in Sub-Saharan African (SSA) countries for waste prevention, few have the proper structure or legislation constructed to encourage it. Re-use entails repeated use of materials in their original form, or transferring them to others, avoiding dumping the items or materials as waste. Recycling happens when people save and transfer reusable items to places where they can be re-created into either similar or new products than throwing them away. Composting is also a method of recycling that mimics nature’s recycling of nutrients and encompasses using decomposer microbes to reprocess yard trimmings, vegetable food scraps, as well as other biodegradable organic wastes [12]. The organic material produced is then added to the soil to fertilise the soil to improve crop yields.

Energy recovery involves two types of recovery methods, namely, combustion of waste materials and anaerobic digestion [13]. However, energy recovery methods may be inappropriate for developing countries for numerous reasons. The air released from the incinerators encompasses a lot of carcinogens. It also entails food pollutants such as volatile organic compounds, heavy metals and dioxins, which require capital investment with little return [10]. Additionally, this technique requires proper landfills for their inert by-products, a factor that increases the cost of the incinerator [14,15]. Disposal denotes to the throwing away of waste, commonly to landfill sites, and poses a great threat to the environment [12]. In SSA countries, current disposal practices such as open dumps, open pits, streams and drainage channels result in human health and environmental problems [16]. In well-developed countries, the landfill gas is connected to the sanitary landfills, but the case may not be the same in low-income countries where the landfill gas is allowed to escape freely to the atmosphere, thus, contributing to global warming [17].

Given the above arguments for a shift in waste management, particularly for the developing countries, with many global trends emerging such as the recognition and need to build low-carbon economies; and to infuse knowledge and innovation throughout economies, urban agriculture may be one of the alternatives concerning managing organic waste. Anastasiou [3] defines urban agriculture as foodstuff and fuel grown within the town or urban areas, produced straight for the marketplace and typically processed and marketed by farmers or their close associates. In this paper, urban agriculture means the production, processing and distribution of food products (both crop and animal) and other products within the urban confines for feeding local populations [18].

In urban agriculture, for example, people use rooftops, backyard gardening, pot culture, and utilising any small pieces of open land spaces for growing food and raising livestock [19]. Generally, with urban agriculture, organic waste material can be utilised in three main ways: fertilising the soil, feeding animals and producing energy [20,21]. MSW usually has high quantities of organic material and nutrients that may be utilised as an input in urban agriculture [18]. Therefore, composting such organic material for use in urban agriculture can be a solution to manage discarded materials in metropolitan regions. Urban families usually depend on the market for vegetables and other food items [22]. Therefore, if the potential of urban areas is appropriately utilised, urban agriculture can aid in waste management and contribute to household food security, human resource utilisation, health hygiene and economic growth in metropolitan communities [23]. Although practising urban agriculture is hypothesised to be beneficial to society, there are some challenges and risks associated with urban agriculture. Some challenges associated with urban agriculture mentioned in the literature include water availability and security issues, health and environmental problems, soil contamination and food safety [24,25]. It is evident that the current ways of managing waste, such as refuse collection and landfilling, are failing to minimise waste in cities [26]. Based on this background, alternative methods of managing organic waste is paramount to attain sustainable cities [27]. Presently, as stated earlier, there is recognition of a shift in global trends with the economic transition taking place in the waste sector, from a collection and disposal dominated waste hierarchy to waste minimisation and recycling [28]. Urban agriculture is postulated to present an opportunity for managing organic waste in cities [29]. However, a knowledge gap exists concerning the production of food in urban ecosystems plus its associated challenges and risks which could potentially result in pollution of the soil, the quality of compost produced and water scarcity [30]. Besides, in developing countries, particularly in South Africa, the concept or urban agriculture is fairly new and relatively few studies have been done concerning the utilisation of organic waste through urban agriculture. This situation warranties further research to help understand the potential challenges and opportunities associated with the utilisation of organic waste through urban agriculture. Again, there are mixed feelings from urban planners concerning the potential for urban agriculture in managing organic waste [21]. The dynamics of urban agriculture still needs to be investigated further if organic waste is to be utilised in urban agriculture [31]. The objective of this paper is to explore the perceived challenges and opportunities of utilising organic waste through urban agriculture in the Durban South Basin using a mixed-methods. A better understanding of the challenges and opportunities associated with the utilisation of organic waste through urban agriculture could be useful in policy-making. This information can be assessed to promote greater integration of urban agriculture with the mainstream urban economy.

## 2. Materials and Methods 

### 2.1. Description and Selection of the Study Area 

The study area where this research took place is the Durban South Basin (29.9206° S, 31.0034° E) in the eThekwini Municipality in the province of KwaZulu-Natal (KZN) in South Africa. A multi-stage sampling approach was applied, whereby eThekwini Municipality in the KZN Province was purposively chosen. The eThekwini Municipality was selected as a case study based on the relevance of the research objectives, data availability and accessibility. Agricultural activities within the eThekwini Municipality include livestock, cropping (vegetable and fruit enterprises), cut flower enterprises, timber and sugarcane production. Very recently, the city of Durban has been crowned the “Greenest City in the World” [32]. The Durban South Basin in eThekwini Municipality was therefore purposively selected based on its development and its urban agricultural potential. The selected areas within the Durban South Basin were: Bluff, Merebank, Wentworth and Umlazi. Generally, these areas exhibit low-quality life, high unemployment levels and pollution. Figure 2 is a map showing the location of the study areas in eThekwini Municipality.

### 2.2. Research Design

This paper presents results emanating from a mixed-method approach to collect primary data. The pragmatism philosophy that caters for both quantitative and qualitative research motivates the choice of a mixed-method approach [33]. A quantitative approach alone does not provide a rich picture of a context. A qualitative approach, on the other hand, is seen as deficient because of the personal interpretations made by the researcher, which may result in biased research results [34]. Generalising findings becomes difficult with a qualitative study because of the limited number of participants studied [34]. A mixed-method approach presents the researcher with a better understanding of the research problem, compared to quantitative or qualitative research methods on their own [33]. In the context of this study, the study needs to understand the perceptions of the challenges and opportunities for utilising organic waste through urban agriculture by households in the Durban South Basin. Perceptions are a qualitative measure yet can be measured and analysed quantitatively. Because waste management through urban agriculture is a relatively new concept in the study areas, a follow up with an exploratory study was deemed appropriate to further understand and substantiate the quantitative study by interviewing officials from the municipality and the agriculture initiative. A mixed-method approach, therefore, was appropriate to advance validity and reliability for this study. Quantitative and qualitative data in a mixed-methods approach was gathered and analysed collectively. This study followed an explanatory sequential design with a quantitative dominance. In an explanatory sequential design, a researcher starts by collecting quantitative data followed by the qualitative data to help to explain or elaborate on the quantitative findings [35]. According to Subedi [36], out of the six mixed-method designs, the explanatory sequential design is highly popular among researchers. The rationale of this approach is that the quantitative data and results provide a general picture of the research problem and more analysis. Through qualitative data collection, this approach is essential in refining, extending or explaining the general views [36]. Figure 3 shows the procedure that was followed in the explanatory sequential design.

#### Conceptual Framework of Organic Waste Utilisation through Urban Agriculture

A conceptual framework helps in highlighting the motives of why a research topic is worth studying [37]. This section presents the conceptual framework of a sustainable model of utilising and managing organic waste through urban agriculture (Figure 4). Globally, urbanisation is increasing. High quantities of organic waste are generated along with MSW in the metropolitan regions. Organic waste material is taken away by being pooled into different waste streams in the sewage systems or dumped as household waste in landfills. Urban agriculture presents an opportunity to use organic waste through a recycling system such as composting. Many advantages are associated with resource recycling. For example, the amount of garbage to be discarded by the municipal waste management unit or authorities can be reduced. Besides, there is also a need to increase food production or food supply in urban areas. The chance of growing and acquiring food that is produced locally is a critical component for some urban dwellers. Therefore, farming near or in the city can contribute significantly to food production [23]. Many urban communities across countries are turning to urban agriculture and gardening. The motive behind this is to increase their ability to acquire healthy food [21]. Urban agriculture can also positively impact on improving the urban environment. For example, urban agriculture benefits the environment, through organic waste reduction, through its utilisation and enhancement of biodiversity [38]. The compost derived from urban organics is incorporated into the soil to increase the amount of nutrients in the soil, which are essential for plant growth. Therefore, nutrient recycling techniques that transmit waste to different urban cultivated areas play a vital role in improving the environment.

Despite the potential benefits that urban agriculture can offer through organic waste utilisation, producing food in the city can have significant challenges and risks. These challenges may include, for example; land and water unavailability issues, lack of a supporting policy from municipal authorities, ease of theft and crop damage by animals, and risk of soil contamination that may need to be managed. Figure 4 shows a conceptual framework of a sustainable model of utilising and managing organic waste through urban agriculture and the associated benefits that can be derived from the practice such as healthy food and a green environment. This framework guided this paper.

### 2.3. Sampling Procedure and Sample Sizes

The target population for this study were household members and key informants within the Durban South Basin of eThekwini Municipality in the areas as mentioned earlier who were practising any form of urban agriculture at the time of the study. Key informants consisted of staff from the Cleaning and Solid Waste Unit of the eThekwini Municipality and officials from an urban agriculture initiative within the Durban South Basin. It was not possible to get a complete sampling frame for all the households practising urban agriculture who reside in the study areas. Therefore, a non-probability (purposive) sampling technique was used to select respondents. The advantage of using purposive sampling is that respondents and study sites that are selected can inform a better understanding of the research problem of the study [39]. A total of two hundred (200) households (that is 50 households were selected from each residential area: Bluff, Merebank, Wentworth and Umlazi) for the quantitative study (survey). These households were selected with the help of the KZN Department of Agriculture in Durban. A sample of 200 respondents was deemed large enough to generate statistically significant results and at the same time, manageable in terms of cost and time. For the qualitative study, ten (10) key informants (that is six (6) staff from the Cleaning and Solid Waste Unit of the eThekwini Municipality and four (4) officials from the urban agricultural initiatives) were interviewed. The Cleaning and Solid Waste Unit of eThekwini Municipality is the major supplier of waste management services for the eThekwini Municipality with a business network of departments that include six units: operational centres, transfer stations, active landfill sites, recycling plants, landfill gas projects and leachate plants [40].

### 2.4. Data Collection and Research Instruments 

Data were collected through a formal survey method (structured questionnaire) for the quantitative approach and key informant interviews for the qualitative approach. As pointed out earlier, data were collected in two phases: firstly, the quantitative approach (survey—questionnaire administration) and secondly, the qualitative approach (interviews administered through key informants). The University of Zululand Research Ethics Committee (UZREC) issued an ethical clearance for this study before its commencement.

#### 2.4.1. The Survey 

Questionnaires were prepared and administered on a house-to-house basis. Data were collected during October/ November 2018 and February 2019, during working hours (08:00–16:00), and the face-to-face interviews, on average, lasted up to 45 minutes. As indicated earlier, the respondents were selected from households who are actively involved in any form of urban agriculture activity. The questionnaires were translated to the local isiZulu language since it is the most spoken language in the study area. Questionnaires were explained to ensure that the respondents understood what was asked of them. Issues covered in the survey were related to the information on the present situation of using organic waste in urban agriculture and the perceived challenges and risks associated with the utilisation of organic waste through urban agriculture by households. The questionnaire primarily used a Likert scale format with responses ranked from 1 to 5: strongly disagree (1), disagree (2), neutral/ indifferent (3), agree (4) and strongly agree (5). Additionally, the perspectives on the potential benefits for organic waste utilisation and management through urban agriculture (such as the production of healthy and fresh crops, organic waste management and nutrient recycling) were assessed.

Data collection was conducted by trained enumerators who were identified before the field survey. The questionnaire was piloted with some respondents who did not form part of the survey. The pilot sample consisted of forty (40) respondents, that is ten (10) from each selected residential area. The purpose of the pilot study was to determine the feasibility of the study as well as to improve its success and effectiveness. Pilot testing helps to improve the validity and reliability of the instrument and to estimate how long it would take to complete the questionnaire.

#### 2.4.2. The Key Informant Interviews 

In-depth interviews (using an interview schedule) with key informants were carried out from January to April 2019. Interviews were conducted during working hours (08:00–16:00), and the discussions on average lasted up to 45 minutes with each key informant. The interviews explored the current practices of organic waste management as well as the perceived potential challenges and benefits of utilising organic waste through urban farming. The interview schedule was designed in English, and the interviews were conducted in English as a medium of communication. The interview process involved the interaction between the researcher and the key informants. Before the actual interview process, the interview schedule was piloted to three (3) municipal officials (staff from the Cleaning and Solid Waste Unit of the eThekwini Municipality) and two (2) officials from the urban agriculture initiative who did not form part of the study. The interviews were tape-recorded, and notes were taken at the same time. Interviews took place inside the interviewees’ offices which were deemed to be essential to ensure that the key informants felt comfortable with their surroundings for the meeting. After the interviews, the researcher engaged with the tape recordings and notes; writing down relevant information. 

### 2.5. Data Management and Analysis

Both quantitative and qualitative data analyses techniques are used in this paper since this study adopted a mixed-methods approach. For quantitative data, raw data was captured and coded on Microsoft Excel 2016 (Microsoft Corporation, Washington, USA) and exported to the Statistical Package for the Social Sciences (SPSS) version 25 (SPSS Inc. (IBM), Chicago, Illinois, USA) software for analysis. Descriptive statistics are used to describe the demographics of respondents, the state of organic waste utilisation through urban farming; the perceived associated challenges and risks, and the potential benefits. Specifically, frequencies and percentages in the form of a Likert scale (5-point) measures and cross-tabulations (Pearson Chi-square correlation) are used. For qualitative data, after transcribing, data was organised with the help of NVivo software, version 10 (QSR International, Melbourne, Australia). Inductive thematic analysis is then applied concerning emerging themes. The thematic analysis highlights pinpointing, examining and recording themes (patterns) within the data [41].

## 3. Results

### 3.1. Demographic Characteristics of the Respondents in the Durban South Basin

The results in Table 1 show that the majority of the respondents (59.5%) in the Durban South Basin among the sampled population were women compared to men (40.5%). The results in this paper show that 40 per cent of the respondents in the Durban South Basin were single, and 41.5, 7.5 and 11 per cent were married, divorced and widowed, respectively (Table 1). The age group distribution of the respondents arranged in categories from youth to adults, show that the elders dominated (43.5%) with the age group of 36 to 50 years, followed by the 50 to 60 and 18 to 35 years age categories, each accounting for 22.5 per cent of the sample. A few respondents (11.5%) were over 60 years old (Table 1). The results in this paper show that about half (51%) of the farmers were unemployed (did not have formal jobs), while a fair share of the respondents (49%) were employed (Table 1). 

Table 1 shows the average monthly household income among the respondents in the Durban South Basin. The results show that a higher proportion of the respondents (40.5%) earned more than US$175.16 per month, 21.5 per cent earned between US$140.11 to US$175.16, 19.5 per cent earned between U$105.07 to US$140.11, 18 per cent earned between US$70.02 to US$105.07 and one respondent earned between US$35.05 to US$70.02. Slightly above half of the respondents (51%) had matriculated, and 27.5 per cent did not have formal education, 12 per cent had a diploma, and 9 per cent had a degree while only one respondent had obtained a postgraduate qualification. In total, the majority of the respondents (72.5%) had some formal training with the highest proportion having a high school education (Table 1). A higher proportion (57%) of the sampled households had family sizes of 3 to 5 members (Table 1). Table 1 summarises the demographic characteristics of the interviewed respondents in the Durban South Basin from the quantitative study.

### 3.2. The Type of Space Utilised for Urban Farming in the Durban South Basin

Table 2 shows that the type of space utilised by urban farmers from the quantitative study for farming purposes includes backyard or courtyard spaces, communal gardens, public or vacant land and rooftops. Most of the respondents (69.5% and 79.5%) affirmed (that is either agreed or strongly agreed) that they were engaged in urban agriculture through backyard gardening and communal gardening, respectively. In contrast, a minority of the respondents (13.5%) in the Durban South Basin were utilising the rooftop method to plant their crops.

### 3.3. The Utilisation of Organic Waste Through Urban Agriculture in the Durban South Basin

Table 3 shows the proportion of the respondents from the quantitative study, who indicated that they utilise organic waste through urban agriculture. Almost all the respondents (95%) indicated that they utilise organic waste in some form in their urban agricultural practices.

#### 3.3.1. Urban Agriculture Initiative within the Durban South Basin

Qualitative data established that there is an urban agriculture initiative within the eThekwini Municipality involved in different projects that are known as the small-scale farmers’ mentorship and community reforestation programmes. The initiative helps farmers about crop cultivation using organic waste as a fertiliser. Qualitative data also shows that the respondents utilise organic waste in growing a range of crops (vegetables and herbs). One of the key informants stated:
“*Urban farmers in eThekwini Municipality cultivate different crops, and they are encouraged to produce a variety of vegetable crops, namely, cabbages, Swiss chard, carrots, brinjal, onions, lettuces, and cucumbers. They also produce herbs such as thyme, parsley and ginger.*”

The key informants were asked to clarify how the agricultural initiative in the eThekwini Municipality works. Figure 5 outlines how the urban agriculture initiative within the Durban South Basin works with small-scale farmers. 

In clarifying how the agricultural initiative in the eThekwini Municipality works, one of the key informants stated:
“*There is a community reforestation programme, which is an initiative that offers training on agroecological farming practices. It consists of about 50 small-scale farmers. Furthermore, the initiative provides mentorship and garden support to small-scale farmers. The programme also helps small-scale farmers to establish nurseries and help to provide linkages to markets. Other programmes that are enhanced by agriculture initiatives educate small-scale farmers about climate-smart agriculture, which enables farmers to grow vegetables under harsh conditions. For example, if there is water scarcity, farmers know how to save water and to irrigate water simultaneously.*”

#### 3.3.2. The Relationship between the Utilisation of Organic Waste through Urban Agriculture and the Demographic Characteristics of the Respondents

A cross-tabulation (Pearson Chi-squared test) was performed between the demographic variables and the utilisation of organic waste through urban agriculture to establish if there were any statistically significant associations. Table 4 shows the demographic variables and their significance (p-values). A cross-tabulation between the utilisation of organic waste through urban agriculture and the gender of respondents, marital status, age, employment status, average household income and household size reveals an insignificant Pearson Chi-square statistic (Table 4). Nonetheless, a cross-tabulation between the level of education of the respondents and the utilisation of organic waste through urban agriculture reveals a Pearson Chi-square statistic (χ2 = 10.135, *p* = 0.038). This finding indicates that a statistically significant association existed between the level of education and the utilisation of organic waste through urban agriculture in the Durban South Basin.

### 3.4. Challenges and Risks Associated with the Utilisation of Organic Waste Through Urban Agriculture

Table 5 and Table 6 present a summary of the challenges and risks associated with the utilisation of organic waste through urban agriculture in the Durban South Basin as perceived by the respondents and key informants from the quantitative and qualitative studies, respectively. 

The results show that the majority of the respondents (89.5%) in the Durban South Basin affirmed (agreed or strongly agreed) that climatic variation was a primary challenge associated with the utilisation of organic waste through urban agriculture (Table 5). A higher proportion of the respondents (86%) in the Durban South Basin also affirmed (agreed or strongly agreed) that lack of land tenure rights was also a challenge associated with the utilisation of organic waste in urban agriculture. This finding is also supported by the qualitative study (Table 6). 

To corroborate the challenge concerning the land issue as revealed by the quantitative study, one of the key informants stated:
“*Land availability is another challenge to urban farmers; this is because the farmers do not have access to land nor have enough capital to buy land. This challenge hinders the farmers’ capacity to expand their farming enterprise.*”

The purpose of urban agriculture policy, whether national, provincial or municipal is to formulate an integrated framework and clear implementation to support urban agriculture. Generally, a policy is a guiding tool for all stakeholders to align and synergise efforts to create an enabling environment. The results show that a higher proportion of the respondents (79.5%) in the Durban South Basin were of the view that there is a lack of a supporting policy concerning the utilisation of organic waste in urban agriculture (Table 5). Again, the majority of the respondents (70%) in the Durban South Basin affirmed that theft and crop damage was a challenge associated with the utilisation of organic waste in urban agriculture (Table 5). This finding on theft and crop damage is also supported by the qualitative study (Table 6). One of the key informants stated:
“*Most urban farmers are experiencing theft daily on their farms. Both animals and human trespassing across the cultivated areas are most likely to cause damage.*” 

The results show that a higher proportion of the respondents (93%) in the Durban South Basin affirmed that there is a risk of soil contamination associated with the utilisation of organic waste in urban agriculture (Table 5). Again, the majority of the respondents (68.5%) consented to the view that there is a risk of food safety associated with the utilisation of organic waste in urban agriculture (Table 5). Both the risks of soil contamination and food safety were also reported from the qualitative study (Table 6). One of the key informants stated:
“*Soil contamination and food safety still pose challenges and risks. Furthermore, lack of education and the know-how for the waste producers to manage such waste without resorting to disposing of is a challenge.*”

The qualitative findings also show that among the challenges of utilising organic waste through urban agriculture in the Durban South Basin, were inadequate access to inputs which were reported to be expensive relative to the selling price (Table 6). One of the key informants stated:
“*Accessing inputs for urban agricultural activities, as well as finding the proper market for selling produce is also a challenge.*”

Water is generally a scarce resource. Acquiring a water usage licence is also problematic in urban areas, especially for farming purposes. The qualitative study established that water scarcity was a challenge associated with urban agriculture (Table 6). One of the key informants stated:
“*Accessing a license to use water is a challenge to urban farmers. Water from the river may be used as a source of water supply for irrigation of the farm crops. The application for the usage of water needs to be submitted to the Department of Water Affairs to attain a certificate to use the water from the river for irrigation; thus a water levy must be then paid monthly. Therefore, some urban farmers do not have enough capital to pay for a water license.*”

The difficulty in segregating waste (inadequate management of organic waste) was also noted to be a challenge associated with urban agriculture (Table 6). One of the key informants stated:
“*Currently, organic waste streams are received in a heterogeneous mix; therefore, the major constraint is the sourcing and separation of organic waste. Depending on the scale at which this is to be carried out, several factors contribute such as the available land on which the urban agriculture is to be carried out, whether this is carried out as an individual or community-based projects as well as linking up the farmer/producer to the supplier.*”

Generally, waste has to be transported from residential areas (households) to landfills. The key informants from the Municipality indicated that limited transportation of organic waste (including the costs of transportation) was a challenge associated with urban agriculture (Table 6). One of the key informants stated:
“*The major challenge that the eThekwini Municipality has is increased transportation requirements for both organic waste and inorganic waste, which in turn contributes towards high transportation costs and limited space for landfills.*” 

Another reported challenge from the municipal side is that of inadequate handling and treatment of a different kind of waste. The key informants indicated that inadequate handling and treatment of organic waste was a challenge associated with urban agriculture (Table 6). One of the key informants stated:
“*eThekwini Municipality collects and processes all types of general solid refuse, including household refuse and business refuse. However, the Municipality does not have any facilities that handle hazardous waste and organic waste.*”

In as much as urban farming can enhance the greening ecology of the city, it is also a potential contributor to environmental pollution. The key informants also indicated that health hazards (such as bad odour) was a challenge associated with urban agriculture (Table 6). One of the key informants stated:
“*The possible risk is the bad odour that will be produced when processing the waste, and it would have bad environmental pollution (air pollution) and a nuisance to other people.*”

#### The Relationship between the Perceived Challenges and Risks and the Utilisation of Organic Waste through Urban Agriculture 

A Pearson Chi-square statistic was performed between the perceived challenges and risks for urban agriculture and the utilisation of organic waste through urban agriculture to establish whether a significant statistical association existed. Table 7 shows the relationship between the perceived challenges and risks for urban farming and the utilisation of organic waste through urban agriculture from the quantitative study.

The results in Table 7 show that a statistically significant association exists between the utilisation of organic waste through urban agriculture and its perceived challenges and risks in the Durban South Basin that include lack of a supporting policy for organic waste utilisation for small-scale urban agriculture (χ2 = 10.264, *p* = 0.016), climatic variation (χ2 = 8.508, *p* = 0.037), lack of land tenure rights (χ2 = 27.463, *p* = 0.000), soil contamination (χ2 = 9.527, *p* = 0.023) and food safety (χ2 = 15.754, *p* = 0.001). However, the perception that there is the ease of theft and crop damage by animals in urban agriculture shows no statistically significant association with the utilisation of organic waste through urban agriculture in the Durban South Basin.

### 3.5. Opportunities for Utilising Organic Waste Through Urban Agriculture in the Duran South Basin

Table 8 and Figure 6 present a summary of the potential benefits associated with the utilisation of organic waste through urban agriculture in the Durban South Basin as perceived by the respondents and key informants from the quantitative and qualitative studies, respectively. About 89 per cent of the respondents in the Durban South Basin were of the view that urban farming creates environmental awareness in urban communities (Table 8). The results show that respondents in the Durban South Basin felt that urban agriculture contributes to the protection of the environment as revealed by the majority of the respondents (87%) (that is, those who agreed and strongly agreed) (Table 8). The majority of the respondents (97.5%) affirmed (agreed or strongly agreed) that practising urban agriculture plays a significant role in food and nutrition security (Table 8).

The perception that practising urban agriculture enhances food security is also supported by the qualitative study (Figure 5). Figure 6 shows an overview of the perceived potential benefits of the utilisation of organic waste through urban agriculture from the qualitative study.

To substantiate the quantitative finding that practising urban agriculture enhances food and nutrition security, one of the key informants stated:
“*The use of organic fertilisers is particularly important in most parts of urban areas, where low availability of nutrients is a serious constraint for food production. Organic fertilisers would improve soil fertility and increase crop yields such as vegetables, thus enhance food security.*”

The majority of the respondents (94% and 81%) in Durban South Basin consented (agreed or strongly agreed) that urban agriculture contributes to the reduction of poverty and helps in job creation, respectively (Table 8). The quantitative results also show that a higher proportion (80.5%) of the respondents in the Durban South Basin were of the view that practising urban agriculture helps in the market expansion for farmers (Table 8). This finding is also supported by the qualitative study (Figure 6). One of the key informants stated:
“*Most urban farmers play a major role in the production of a variety of vegetable crops and herbs. The crops produced are primarily for household consumption; the remaining produce and herbs are sold to the market.*”


The qualitative findings also show that improved soil nutrients are one of the perceived benefits associated with the utilisation of organic waste through urban agriculture in the Durban South Basin (Figure 6). One of the key informants stated:
“*The utilisation of organic waste, leads to the production of healthy crops in urban agriculture, which in turn contributes to crops being better positioned to tolerate pests and diseases. Therefore, it needs proper management so that the application of it, correct amounts and timing, and by methods that are appropriate to agronomic and environmental requirements are essential for fertilisation of the soil.*”

The key informants also indicated that improved productivity is also a perceived benefit associated with the utilisation of organic waste (Figure 6). One of the key informants stated:
“*Most of the urban farmers use organic waste material to make compost; therefore, the compost is utilised as a fertiliser to the soil for good produce. Compost provides the ability for soil to sustain agricultural plant growth which leads to the consistent yields of high quality.*”

#### The Relationship between the Perceived Benefits for Urban Farming and the Utilisation of Organic Waste through Urban Agriculture 

A Pearson Chi-square statistic was performed between the perceived benefits to urban agriculture and the utilisation of organic waste through urban agriculture to establish whether any statistically significant association exists. Table 9 shows the relationship between the perceived benefits for urban farming and the utilisation of organic waste through urban agriculture from the quantitative study. 

The results in Table 9 show that there is no statistically significant association between the utilisation of organic waste through urban agriculture and its perceived potential benefits in the Durban South Basin which include raising awareness of environmental issues (χ2 = 2.121, *p* = 0.548), protection of the environment (χ2 = 2.301, *p* = 0.512), combating poverty (χ2 = 6.079, *p* = 0.108), economic savings on food (χ2 = 6.859, *p* = 0.077), job creation (χ2 = 5.026, *p* = 0.285) and market expansion for farmers (χ2 = 3.326, *p* = 0.344). However, the results show that a statistically significant association exists between the utilisation of organic waste and the perception that utilising organic waste through urban agriculture enhances food and nutrition security (χ2 = 17.442, *p* = 0.001) and unveils opportunities for business incubation (χ2 = 18.994, *p* = 0.001).

## 4. Discussion

The cross-tabulation analysis in the demographic section reveals that there is no statistically significant relationship between the use of organic waste and the socio-demographic variables apart from the level of education of the respondents in the Durban South Basin. This finding could be attributed to almost all the respondents indicating that they use some form of organic waste in their agricultural practices. While the literature notes that these variables influence participation in urban agriculture, this paper establishes that concerning the use of organic waste specifically, these factors have less of an influence in the Durban South Basin.

Although producing food in the city is hypothesised to be beneficial to society, there are some challenges and risks associated with urban agriculture [42]. Several challenges and risks associated with the utilisation of organic waste in urban agriculture in the Durban South Basin were noted as perceived by the respondents and key informants. Respondents affirmed the lack of a supporting policy concerning the utilisation of organic waste in urban agriculture. Further, the results show that a statistically significant association exists between the utilisation of organic waste through urban agriculture and the lack of supporting policy to urban farmers in the Durban South Basin. Although this paper cannot generalise due to the purposive nature of the sampling, a similar view was reported by Hunold et al. [43] that in most cases, urban farming policies are not inclusive of small-scale farmers in urban areas; instead, there is a tendency to focus primarily on commercial farmers who can produce and sell their produce to the market. To the researchers’ knowledge, there is currently no urban agriculture policy in South Africa except for the cities of Cape Town and Johannesburg. Without adequate support and a guiding framework, this situation could potentially discourage small-scale farmers from participating in urban farming activities, including the utilisation of organic waste. Urban agriculture policy is, therefore, necessary to develop a common vision for urban agriculture for the eThekwini Municipality and its residents, establishing and clarifying the role of key stakeholders, formulating strategic objectives to guide the implementation of organic waste utilisation through urban agriculture. Such a policy could also guide consultative forums for stakeholder participation, institutional frameworks and assistance programmes to develop urban agriculture and organic waste management. There is a need to develop an agricultural policy that focuses on the current approaches for utilising organic waste through urban agriculture, supported by all stakeholders. 

Wortman and Lovell [31] state that the climatic and atmospheric variations in urban areas create challenges to urban farmers. Our finding confirms this assertion that climatic variation is a challenge associated with the utilisation of organic waste in urban agriculture. Additionally, a statistically significant association was established between the utilisation of organic waste through urban agriculture and climatic variation in the Durban South Basin. Although the nature of the study limits us to generalise; the finding is similar to the study of Malhotra [44] who revealed that climatic variation, for example, high temperatures during the day or night, hinders photosynthetic processes that result in a reduction of crop yields. Although the climate issue is a general challenge in farming, a recent study by Searchinger et al. [45] suggests that organically produced food may have more impact to climate change than conventionally grown food as previously known. They explained that the low yield per hectare required to grow the same amount of food using fertilisers would require more land-use, thus indirectly contributing to higher carbon dioxide emissions. Generally, temperatures are often high in most urban areas, when the vapour pressure is exceptionally high, plants consume more water, thus creating moisture stress, and the absence of moisture threatens crop growth. Currently, with no policy integrated into a food or climate strategy, it is difficult to clinch how organic waste utilisation through urban agriculture can strive as a viable waste management strategy. 

Urban city planners are gradually more concerned about solving the issue of access to land to be utilised for agricultural purposes [46]. Lack of land tenure rights was a reported challenge associated with the utilisation of organic waste in urban agriculture in the Durban South Basin. Also, a statistically significant association existed between the utilisation of organic waste through urban agriculture and the lack of land tenure rights. Although this study cannot generalise, a similar view was reported by Cerrada-Serra et al. [47] that urban farmers are constrained by the inability to expand their farms due to lack of land tenure rights. Urban farmers are required first to consult municipal authorities or landowners to get approval whether they can make use of any plots. This situation could discourage farmers or households who wish to utilise organic waste through urban farming. An urban agricultural policy already alluded to, could also potentially focus on improving the poor land distribution and tenure of land. Availability of land, plus tenure rights may encourage households to participate in urban agricultural activities; thus, the utilisation of organic waste through urban agriculture.

In urban areas, crop theft is growing at an alarming rate with the damage and the loss of crops at night [48]. Quantitative results revealed that most farmers utilised any open spaces or land within the city for farming. Most of these spaces are not fenced or protected. This situation could potentially expose crops to theft or damage by animals. Most respondents were convinced that theft and crop damage was a challenge associated with the utilisation of organic waste in urban agriculture. However, the perception that there is the ease of theft and crop damage by animals in urban agriculture shows a weak association with the utilisation of organic waste through urban agriculture in the Durban South Basin.

Although organic waste contains numerous nutrient-rich materials, the major problem is that compost can be severely contaminated. Arguably, composting is a safe urban agriculture practice, but can be contaminated with heavy metals because of lack of sorting and handling skills of inorganic waste by households which may greatly contaminate the soil. A higher proportion of the respondents agreed that there is a risk of soil contamination associated with the utilisation of organic waste in urban agriculture. The cross-tabulation results also show that a statistically significant association existed between the utilisation of organic waste through urban agriculture and soil contamination in the Durban South Basin. Vegetables that are cultivated by urban farmers in open spaces near the roads and industries are also exposed to soil contamination risks if crops absorb nutrients from contaminated soil, people who consume those crops may be exposed to health risks [49]. This perception could deprive urban farmers who wish to utilise organic waste through urban farming, but concerned about health risks.

Respondents agreed with the notion that food safety was a risk associated with the utilisation of organic waste in urban agriculture in the Durban South Basin. A statistically significant association was established between the utilisation of organic waste through urban agriculture and the risk of food safety. Wortman and Lovell [31] stated that some water-based vegetables and fruits like spinach, lettuce, carrots, cucumbers, oranges and grapes could be affected by *Escherichia coli (E.coli)* if present in the organic waste (perhaps even natural fertilisers such as cow dung). If people eat food that is poisoned by *E.coli*, this could result in infection. This perception could discourage farmers or households who wish to utilise organic waste through urban farming.

Additionally, qualitative findings also reveal similar and other challenges and risks associated with the utilisation of organic waste through urban farming, namely; land unavailability, theft, difficulty in segregating waste, difficulty in accessing inputs, water scarcity, limited transportation of organic waste, inadequate handling of organic waste, soil contamination, food safety and being a health hazard (bad odour). The knowledge gained here contributes to the alertness of environmental problems and morals associated with the use of organic waste through urban agriculture.

Although the effect of urban agriculture to the economy remains unquantified, its practice is commonly postulated to play a significant role in society, for example, that it offers benefits and opportunities to urban dwellers [50]. The results confirm the view that urban farming creates environmental awareness in urban communities. Although we cannot take a broad view, this finding is supported by the study of Skar et al. [51] who reveal that the combination of communal, rooftop and backyard gardens contribute to the greening of metropolitan cities and they offer several environmental benefits through, for example, conservation of groundwater as well as vegetation carbon restoration as means to preserve the environment.

There is growing literature that shows a shifting trend of food insecurity from rural to urban areas. This situation is so because urban residents primarily depend on food imports and buying. The literature on urban farming hypothesises that food grown within the city can help minimise this challenge. Although we cannot oversimplify our results, the results in this paper confirm the findings from the literature that show that urban agriculture can play a significant role in food and nutrition security. Generally, the respondents agreed with the view that urban farming enhances food and nutrition security. A statistically significant association existed between the utilisation of organic waste and the perception that utilising organic waste through urban agriculture enhances food and nutrition security in the Durban South Basin. The results are similar to the previous studies of Abdel-Shafy et al. [46], Muller et al. [52] and Knapp and van der Heijden [53] which reveal that converting organic waste into fertiliser through urban agriculture promotes the cultivation of nutritious food and improved yield. However, some literature, for example, Corrigan [54] and Mkwambisi et al. [55] critique urban agriculture as unable to distribute all the required nutritional needs of societies.

It is evident from the results that practising urban agriculture helps in economic savings on food. Despite the purposive sampling nature of the study, we find the results to be consistent with the previous study of Glasser [56] which reveal that both communal and backyard gardening in the cities can offer vegetables and other cash crops at a lower price than those obtained at commercial markets. Nonetheless, a statistically significant association between the utilisation of organic waste through urban agriculture and the perceived benefit of economic savings on food could not be established.

Unemployment is widespread in South Africa with more than half of the youth unemployed. Agriculture is reported to contribute to a significant proportion of employment in rural areas. From the literature, urban farming is also postulated to be a potential creator of employment opportunities. The results support the perspective that practising urban agriculture can help in job creation. This finding is consistent with the previous study of Rahman et al. [57] who report that many urban agricultural projects, especially those specialising in converting organic waste to increase the soil nutrients and managing waste, play a significant role in creating job opportunities. Such urban agricultural projects have developed programmes to provide skills, knowledge and job training for the youth. Nonetheless, a statistically significant association between the utilisation of organic waste through urban agriculture and the perceived benefit of job creation in the Durban South Basin could not be established.

With the growing unemployment in South Africa, the entrepreneurial agenda is on the rise. Urban farming and the use (recycling) of organic waste could offer such opportunities. Most of the respondents held the view that utilising organic waste through urban agriculture opened doors for business incubation. A statistically significant association existed between the utilisation of organic waste and the perception that utilising organic waste through urban agriculture unveils opportunities for business incubation in the Durban South Basin. This finding cannot be generalised; however, similar previous studies support this assertion, for example, by Frankelius et al. [58] who revealed that transforming waste into organic fertiliser through urban agriculture offers the potential for entrepreneurship, thus creating the most profitable agricultural-related business. However, Hunold et al. [43] found that if the opportunity cost of land and labour are accounted for, urban farming, especially for smaller farms without external funding, is not financially sustainable.

Generally, small-scale farmers do not grow for the market but household consumption. Those who produce in surplus usually struggle to access markets, perhaps because of not meeting quality standards and not being able to supply the market consistently. Again, utilising organic waste for food production and marketing may potentially discourage customers; especially those who hold the view that urban agriculture is a risk to food safety. However, most of the respondents in the Durban South Basin were convinced that utilising organic waste through urban agriculture can help in market expansion for the farmers. Nowadays, there is a rise in health-conscious consumers; those who believe that the use of inorganic synthetic fertilisers is not good for food production and healthy eating. Such consumers may guarantee a market for organically produced food. Nonetheless, we could not establish a statistically significant association between the utilisation of organic waste through urban agriculture and the perceived benefit of market expansion for farmers in the Durban South Basin.

The results from the qualitative findings generally corroborate the perceived benefits from the utilisation of organic waste through urban agriculture from the quantitative study. These include improved food security, enhanced market expansion for farmers, improved soil nutrients and productivity. Overall, the results suggest that utilising organic waste through urban agriculture has considerable potential benefits to offer. Besides, there is an opportunity to manage organic waste in the Durban South Basin, which is becoming a pertinent issue in most metropolitan areas if the potential challenges and risks can be minimised.

## 5. Conclusions

There are relatively few studies which have been conducted to investigate the potential challenges and benefits for organic waste utilisation through urban agriculture, particularly in South Africa. The results of this paper, although they do not deviate from other literature can be of great importance to policymakers, researchers, urban agriculture initiatives and relevant stakeholders interested in organic waste management in the eThekwini municipality, and similar contexts nationally and globally. This paper explored the perceptions of the challenges and benefits for organic waste utilisation through urban agriculture in the Durban South Basin using a mixed-method approach. The results show a weak association of the influence of demographics on the utilisation of organic waste through urban agriculture except for education. Findings reveal some important potential perceived challenges and risks associated with the utilisation of organic waste in urban farming namely; lack of a supporting policy, climatic variation, lack of land tenure rights, soil contamination and food safety concerns, the difficulty in segregating waste by households, water scarcity, difficulty in accessing inputs, limited transportation of organic waste by the municipality, inadequate handling and treatment of organic waste, and bad odour (health hazard). Besides the benefit to improve the environmental quality of cities through waste reduction, organic waste utilisation through urban agriculture is also perceived to offer urban dwellers opportunities for sustainable healthy and fresh food production, and some economic benefits. Some recommendations, although not exhaustive, are put forward:-There is a need for an integrated policy linked to food production and climate mitigation and adaptation strategies that will regulate the management of organic waste in urban agriculture by all stakeholders.-Along with the climate strategy, urban farmers should be educated to grow alternative organic food crops that are climate-smart.-Urban development projects, guided by an urban food production policy, should make reservations of space for urban agriculture. A co-operation between the municipal officials and the Department of Agriculture, Rural Development and Land Reform should also facilitate and address the land distribution and tenure rights for urban farmers.-Composting and vermiculture are some safe examples of organic waste practices that can be explored by farmers. Additionally, urban farmers can also explore the use of rooftops (designed to be conducive for the cultivation of crops). The rooftop method minimises ground soil pollution.-Urban farmers should cultivate crops at least 30 to 100 metres away from roads and industries to avoid contamination.-There is a need to thoroughly educate urban farmers on how to recycle, manage organic waste and properly separate it from inorganic waste. Besides proper waste management education, farmers can be supplied with composting bags.-Municipal officials should assist urban farmers in applying for water licenses from the Department of Water Affairs for access to use water from the nearby rivers. Additionally, the government can subsidise the water usage costs for urban farmers. Water harvesting techniques, for example, from rooftops, can also be explored.-Urban agriculture initiatives, working with extension officers can help urban farmers to obtain inputs, for example, seeds at affordable subsidised prices.-The municipality should erect numerous waste handling facilities around the city to minimise the transportation requirements and costs.-The municipality and urban agriculture initiatives can assist urban farmers with organic waste management technologies that are environmentally friendly, for example, biotechnological treatment of organic waste.

This investigation has not reached a dead end. Further research needs to be undertaken for effective, economic and sustainable utilisation of organic waste through urban agriculture. Extended and future research may focus on characterising and establishing the organic waste volumes in cities, quantifying its contribution to the mainstream urban economies, the role of policy in urban agriculture (inclusive or organic waste utilisation), including the roles and responsibilities of municipal authorities and the private sector in organic waste management. Increased sample size and extending the research to other metropolitan areas in South Africa can be explored to enhance the efficacy of the current findings. 

## Figures and Tables

**Figure 1 ijerph-17-01158-f001:**
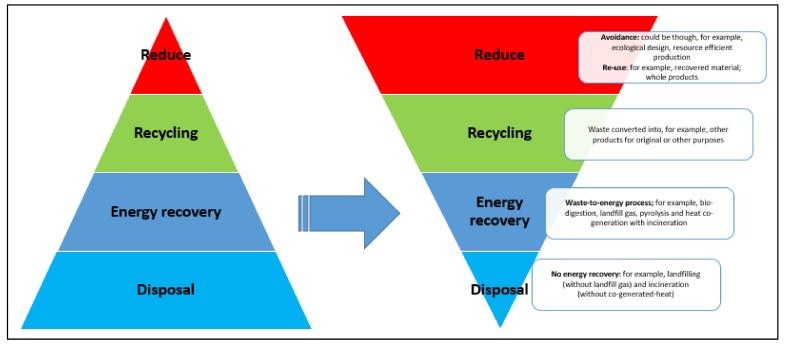
The waste management hierarchy.

**Figure 2 ijerph-17-01158-f002:**
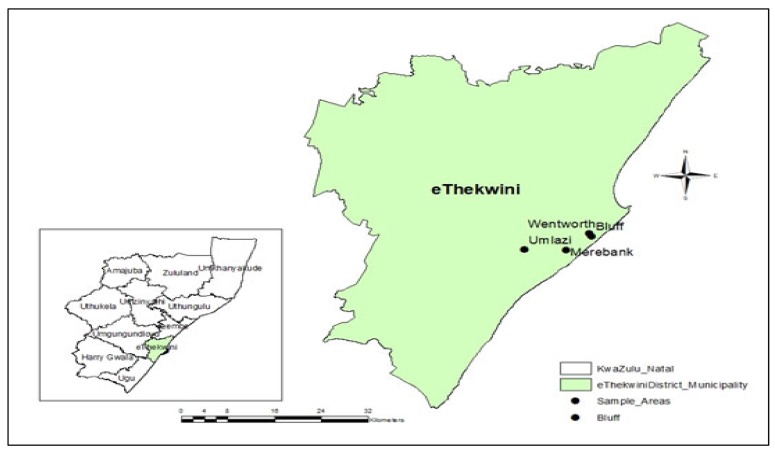
Map showing the study area in eThekwini Municipality in KwaZulu-Natal.

**Figure 3 ijerph-17-01158-f003:**
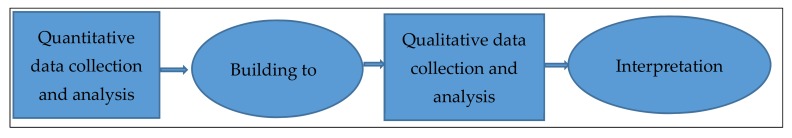
An outline of the explanatory sequential design undertaken in this study.

**Figure 4 ijerph-17-01158-f004:**
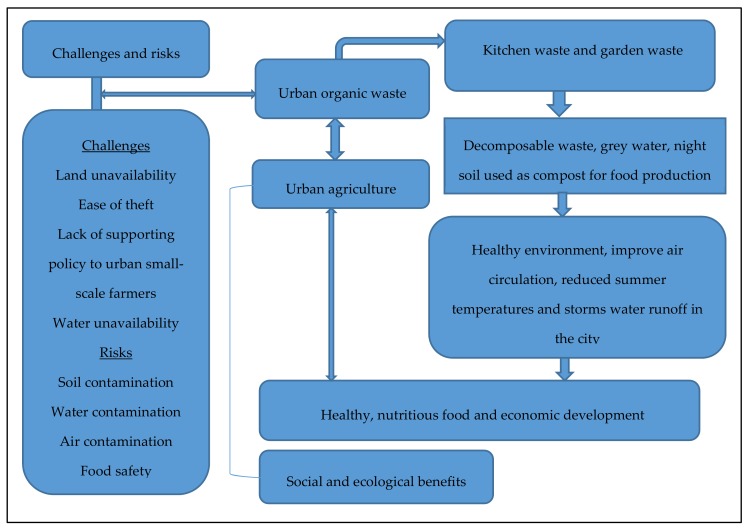
Conceptual framework of a sustainable model of utilising and managing waste through urban agriculture.

**Figure 5 ijerph-17-01158-f005:**
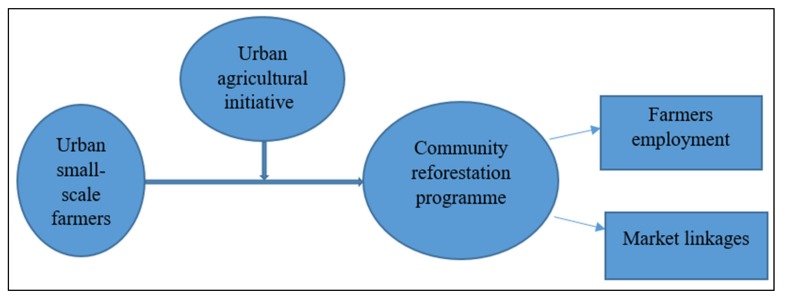
An outline of how the urban agriculture initiative within the Durban South Basin works.

**Figure 6 ijerph-17-01158-f006:**
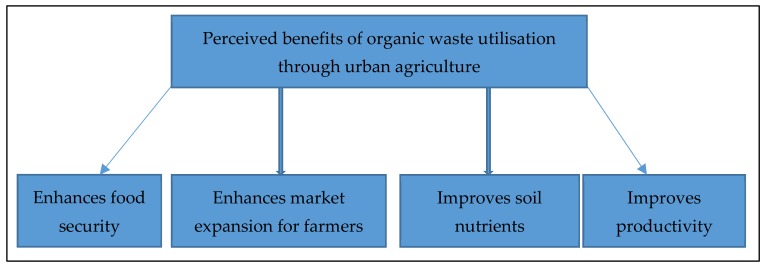
Perceived benefits of organic waste utilisation through urban agriculture.

**Table 1 ijerph-17-01158-t001:** The demographic characteristics of the respondents in the Durban South Basin (n = 200).

Demographic Variable	Frequency	Percentage (%)
**Gender** Female Male Total	119 81 200	59.5 40.5 100.0
**Marital status** Single Married Divorced Widowed Total	80 83 15 22 200	40.0 41.5 7.5 11.0 100.0
**Age (years)** 18–35 36–50 50–60 >60 Total	45 87 45 23 200	22.5 43.5 22.5 11.5 100.0
**Employment status** Unemployed Employed Total	102 98 200	51.0 49.0 100.0
**Monthly household income (USD)** 35.05–70.02 70.09–105.07 105.07–140.11 140.11–175.16 >175.16 Total	1 36 39 43 81 200	0.5 18.0 19.5 21.5 40.5 100.0
**Education** None Matric Diploma Degree Postgraduate Total	55 102 24 18 1 200	27.5 51.0 12.0 9.0 0.5 100.0
**Household size** <3 3–5 6–8 >8 Total	12 114 65 9 200	6.0 57.0 32.5 4.5 100.0

**Table 2 ijerph-17-01158-t002:** The type of space utilised for urban farming in the Durban South Basin (n = 200).

Type of Space Utilised for Urban Farming Purposes	Level of Agreement/Disagreement (%)						
	SD (1)	D (2)	N (3)	A (4)	SA (5)	$\bar{x}$	σ
Backyard or courtyard	25.5	5.0	-	27.0	42.5	3.56	1.655
Communal garden	16.0	4.0	0.5	39.5	40.0	3.84	1.410
Public land or vacant land	21.5	20.0	0.5	21.0	37.0	3.32	1.631
Rooftops	52.0	34.5	-	8.0	5.5	1.81	1.142

SD; D; N; A; SA; $\bar{x}$; σ denotes strongly disagree; disagree; neither; agree; strongly agree; sample mean and standard deviation.

**Table 3 ijerph-17-01158-t003:** The proportion of respondents who utilise organic waste through urban agriculture in the Durban South Basin (n = 200).

Do You Utilise Organic Waste Through Urban Agriculture?	Frequency	Percentage (%)
No	10	5.0
Yes	190	95.0
**Total**	**200**	**100.0**

**Table 4 ijerph-17-01158-t004:** The relationship between demographic variables and the utilisation of organic waste through urban agriculture in the Durban South Basin (n = 200).

Demographic Variable	Pearson Chi-Square	*p*-Value
Gender	0.004	0.947
Marital status	2.187	0.534
Age	5.520	0.137
Employment status	1.948	0.163
Average income	4.953	0.292
Level of education	10.135 **	0.038
Household size	3.541	0.896

** denotes statistical significance at the 0.05 level (2-tailed).

**Table 5 ijerph-17-01158-t005:** The challenges and risks associated with the utilisation of organic waste through urban agriculture in the Durban South Basin as perceived by the respondents (n = 200).

Challenges and Risks Associated with the Utilisation of Organic Waste Through Urban Agriculture	Level of Agreement/Disagreement (%)						
	SD (1)	D (2)	N (3)	A (4)	SA (5)	$\bar{x}$	Σ
Perceived challenges							
Lack of a supporting policy for small-scale urban agriculture practitioners	-	5.0	15.5	48.5	31.0	4.06	0.816
Climatic variation	-	6.5	4.0	39.0	50.5	4.34	0.834
Lack of land tenure rights	0.5	5.0	8.5	53.5	32.5	4.13	0.802
Ease of theft and crop damage by animals	0.5	13.0	16.5	42.0	28.0	3.84	0.995
Perceived risks							
Soil contamination	-	4.0	3.0	40.0	53.0	4.42	0.739
Food safety	-	9.5	22.0	47.0	21.5	3.81	0.884

SD; D; N; A; SA; $\bar{x}$; σ denotes strongly disagree; disagree; neither; agree; strongly agree; sample mean & standard deviation.

**Table 6 ijerph-17-01158-t006:** A summary of the challenges and potential risks of organic usage in urban agriculture from the qualitative study.

Challenge	Potential Risk
Land unavailability	Soil contamination
Theft	Food safety
Difficulty in segregating waste (inadequate management)	Health hazard (bad odour)
Difficulty in accessing inputs	
Water scarcity	
Limited transportation of organic waste	
Inadequate handling and treatment of organic waste	

**Table 7 ijerph-17-01158-t007:** The relationship between the perceived challenges and risks for urban farming and the utilisation of organic waste through urban agriculture in the Durban South Basin (n = 200).

Perceived Challenges And Risks For Urban Agriculture	Pearson Chi-Square	*p*-Value
Lack of a supporting policy for small-scale urban agriculture	10.264 **	0.016
Climatic variation	8.508 **	0.037
Lack of land tenure rights	27.463 ***	0.000
Ease of theft and crop damage by animals	1.906	0.753
Soil contamination	9.527 **	0.023
Food safety	15.754 ***	0.001

***; ** denotes statistical significance at the 0.01 and 0.05 levels, respectively (2-tailed).

**Table 8 ijerph-17-01158-t008:** Opportunities for utilising organic waste through urban agriculture in the Durban South Basin (n = 200).

Potential for Organic Waste Utilisation and Management Through Urban Agriculture	Level of Agreement/Disagreement (%)						
	SD (1)	D (2)	N (3)	A (4)	SA (5)	$\bar{x}$	Σ
Awareness of environmental issues	-	4.0	7.0	40.0	49.0	4.34	0.779
Protection of environment	-	4.5	8.5	46.5	40.5	4.23	0.788
Enhancing food and nutrition security	-	0.5	2.0	38.0	59.5	4.57	0.583
Combating poverty	-	1.5	4.5	52.0	42.0	4.35	0.639
Economic savings on food	-	3.5	11.0	44.5	41.0	4.23	0.781
Job creation	0.5	8.0	11.5	42.5	37.5	4.09	0.923
Business incubation	-	0.5	3.0	43.5	53.0	4.20	0.849
Market expansion for farmers	-	5.5	14.0	42.0	38.5	4.14	0.855

SD; D; N; A; SA; $\bar{x}$; σ denotes strongly disagree; disagree; neither; agree; strongly agree; sample mean & standard deviation.

**Table 9 ijerph-17-01158-t009:** Relationship between the perceived benefits for urban farming and the utilisation of organic waste through urban agriculture in the Durban South Basin (n = 200).

Perceived Benefits To Urban Agriculture	Pearson Chi-Square	*p*-Value
Creating awareness of environmental issues	2.121	0.548
Protection of the environment	2.301	0.512
Enhancing food and nutrition security	17.442 ***	0.001
Combating poverty	6.079	0.108
Economic savings on food	6.859	0.077
Job creation	5.026	0.285
Business incubation	18.994 ***	0.001
Market expansion for farmers	3.326	0.344

*** denotes statistical significance at the 0.01 level (2-tailed).

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
