# Peer review of "Perceptions of the Challenges and Opportunities of Utilising Organic Waste through Urban Agriculture in the Durban South Basin"

_ijerph, 2020, doi:10.3390/ijerph17041158_

Round 1

Reviewer 1 Report

I have gone through the manuscript entitled " Perceptions of the challenges and opportunities of utilizing organic waste through urban agriculture in the Durban South Basin". This study analyzed the perceived potential challenges and opportunities for organic waste utilization and management through urban agriculture in the KwaZulu-Natal Province of South Africa. The topic could belong the areas of interest to the International Journal of Environmental Research and Public Health audience. The present manuscript could be considered to publish in the journal after a major revision.

I would give the specific comments below that could help the authors improve their manuscript.

# Abstract:

*The abstract should be rewritten by detailing the aim and concept of the manuscript. The abstract should state briefly the purpose of the research, the principal results and major conclusions.

* The abstract should be revised with the benefits of the study findings and recommendations as a way forward. 

* Provide significant words which are more relevant to the work in logical sequence as ‘keywords’. Also use keywords which are not present in title.

# Introduction:

* The introduction section is required to be improved. The present introduction is very general and need to be elaborative to explore the actual philosophy to design the study. The introduction is insufficient to provide the state of the art in the topic. Hypothesis should be given. How this work is different from the available data?

The originality and novelty of the paper need to be further clarified. What progress against the most recent state-of-the-art similar studies was made in this study?

*The introduction of the paper must be extended and reformulated in order to provide a more comprehensive approach.

* The objectives of the study are missing in the last paragraph of introduction. Pls. elaborate.

# Materials and methods:

*Study area; provide coordinates.

# Results and discussion:

*The manuscript does not provide interesting and technically sound discussion; it would be better to use more recent references in discussion.

*Authors have presented their result but these results obtained, authors did not justify in the discussion.

*Under section, discussion, it is recommended to discuss and explain what the appropriate policies should be based on the findings of this study. Also, the results should be further elaborated to show how they could be used for real applications. 

# Conclusion

Pls. conclude with more focus on the major outcomes of the paper and future perspectives

Author Response

.

Reviewer 2 Report

The article studies the challenges and opportunities of using organic wastes in urban agriculture practices. The challenges and opportunities are studied based on the understating of interviewees through a mix of quantitative and qualitative approaches. My suggestions for improvement of the text are:

L 68-78: This study is focused on only urban agriculture practices in the Durban South Basin. It would be nice if the introduction includes some background information about the context of the study.  

L 97-99: In this part of research design you are underlying the limitations of quantitative and qualitative research methods. However, both methods if applied correctly can provide valuable data. So, besides general (potential) limitations of the methods, it would be good to explain why a mix method is the correct choice for your study.

L 155-157: The reader may face following questions: What was target population size? Why these four residential areas?

L 173-177: Would be useful to have information about: How did you find these 200 households? How did you approach them (Online or you met them in the urban farms)?

L518-521: You mention ‘’The majority of the respondents agreed that there is a lack of a supporting policy concerning the utilisation of organic waste in urban agriculture’’. My question is: are there any local or national supporting policies? If there are no supporting policy, how does the respondent’s perception will add to the current knowledge?

L 522-524: ‘’Further, the results show that a statistically significant association exists between the utilisation of organic waste through urban agriculture and the lack of supporting policy to urban farmers in the Durban South Basin’’ Can you elaborate on this? What kind of supporting policies are needed? How lack of policies is creating challenges for the use of organic waste through urban agriculture?

L540-541: Isn’t the issue of temperature more a general challenge for urban agriculture? How is this directly connected to the use of organic waste?

L510-603: In the most paragraphs of your discussion this repetition can be seen: ‘’The findings show that a higher proportion of the respondents in the Durban South Basin agreed ...’’ and ‘’This paper established that a statistically significant association exists between the utilisation of organic waste through urban agriculture and …’’, is this repetition needed?

Author Response

.

Round 2

Reviewer 1 Report

The authors have addressed all the comments. Hence, the paper may be accepted in its current form